# Natural Cross-Kingdom Spread of Apple Scar Skin Viroid from Apple Trees to Fungi

**DOI:** 10.3390/cells11223686

**Published:** 2022-11-20

**Authors:** Mengyuan Tian, Shuang Wei, Ruiling Bian, Jingxian Luo, Haris Ahmed Khan, Huanhuan Tai, Hideki Kondo, Ahmed Hadidi, Ida Bagus Andika, Liying Sun

**Affiliations:** 1State Key Laboratory of Crop Stress Biology for Arid Areas and College of Plant Protection, Northwest A&F University, Yangling 712100, China; 2College of Agronomy, Northwest A&F University, Yangling 712100, China; 3Institute of Plant Science and Resources, Okayama University, Kurashiki 710-0046, Japan; 4U.S. Department of Agriculture, Agricultural Research Service, Beltsville, MD 20705, USA; 5College of Plant Health and Medicine, Qingdao Agricultural University, Qingdao 266109, China

**Keywords:** Viroid, filamentous fungi, cross-infection, hypovirulence, Mycobiome

## Abstract

Viroids are the smallest known infectious agents that are thought to only infect plants. Here, we reveal that several species of plant pathogenic fungi that were isolated from apple trees infected with apple scar skin viroid (ASSVd) carried ASSVd naturally. This finding indicates the spread of viroids to fungi under natural conditions and further suggests the possible existence of mycoviroids in nature. A total of 117 fungal isolates were isolated from ASSVd-infected apple trees, with the majority (85.5%) being an ascomycete *Alternaria alternata* and the remaining isolates being other plant-pathogenic or -endophytic fungi. Out of the examined samples, viroids were detected in 81 isolates (69.2%) including *A. alternata* as well as other fungal species. The phenotypic comparison of ASSVd-free specimens developed by single-spore isolation and ASSVd-infected fungal isogenic lines showed that ASSVd affected the growth and pathogenicity of certain fungal species. ASSVd confers hypovirulence on ascomycete *Epicoccum nigrum*. The mycobiome analysis of apple tree-associated fungi showed that ASSVd infection did not generally affect the diversity and structure of fungal communities but specifically increased the abundance of *Alternaria* species. Taken together, these data reveal the occurrence of the natural spread of viroids to plants; additionally, as an integral component of the ecosystem, viroids may affect the abundance of certain fungal species in plants. Moreover, this study provides further evidence that viroid infection could induce symptoms in certain filamentous fungi.

## 1. Introduction

Viroids are the smallest known infectious non-coding, circular, single-stranded RNA molecules (234–401 nucleotides, nt) that naturally infect many plant hosts, replicate autonomously, and cause important diseases [1,2]. Since viroids are nonprotein-coding RNAs, they must be replicated by preexisting cellular RNA polymerases and processing enzymes [3,4,5]. Moreover, one step of the replication cycle in the family *Avsunviroidae* is catalyzed by hammerhead ribozymes (ribonucleic acid enzyme) embedded in viroid strands, which allows for self-cleavage and ligation to form circular RNAs [3,5,6].

Viroids, as subviral agents, are classified into 2 families, 8 genera, and 32 species [2]. The two families are *Pospiviroidae* and *Avsunviroidae* [7,8]. The replication of the members of the family *Pospiviroidae* takes place in the nucleus, while members of the family *Avsunviroidae* replicate in the chloroplasts [4,9,10].

Viroids have been reported to infect and replicate in susceptible plant species [2,11,12], the unicellular yeast fungus *Saccharomyces cerevisiae* [13], cyanobacteria [14], filamentous plant-pathogenic fungi [15], and to some extent, an oomycete *Phytoptora infestant* [16]. Previously, our research group artificially introduced viroids into three filamentous plant pathogenic ascomycete fungi, *Cryphonectria parasitica*, *Valsa mali*, and *Fusarium graminearum* via the transfection of fungal spheroplasts with viroid RNA. Our inoculation experiments showed that ASSVd, chrysanthemum stunt viroid, peach latent mosaic viroid, potato spindle tuber viroid (PSTVd), and citrus exocortis viroid replicated initially in fungi but were eliminated after multiple subsequent sub-culturing steps, whereas hop stunt viroid (HSVd), ASBVd, iresine viroid 1, and avocado sunblotch viroid stably infected at least one of these fungal hosts [15]. We further demonstrated that HSVd could be bi-directionally transmitted between plant and fungus during the fungal colonization of the plant. Similarly, the transmission of PSTVd between *P. infestans* and the host plants under laboratory conditions was also recently demonstrated [16]. However, the occurrence of the cross-kingdom infection of viroids between plants and fungi in natural settings is still unexplored.

Plants can host various organisms, including fungi, bacteria, phytoplasmas, viruses, nematodes, algae, and protozoa; many of these are parasitic and cause plant diseases, whereas some carry out essential functions necessary for plant growth and survival under stress conditions [17,18]. Fungal colonization/associations (mycobiome or fungal microbiome) are common in land plants [19]. Studies of plant-associated fungal communities have been focused on the rhizosphere (the root-soil interface) and the phyllosphere (the surface and interior of the aerial region of the plant) with mutualistic, pathogenic or commensal relationships [20,21,22]. Mainly plant pathogenic fungi have been investigated; they commonly phyllosphere-colonize their host plants and cause many devastating plant diseases, such as blights, smuts, rusts, and powdery mildew [23]. Studies revealed that environmental conditions and host species influence the structure of the phyllosphere fungal community [17,24,25]. To date, however, the influence of viruses and viroids on the phyllosphere microbiome under natural conditions remains unexplored. Although the effects of viruses and viroids on the host plants are well known, the impact of these infections on the fungus-plant ecosystem is uncertain and merits further investigation.

ASSVd, the type species of the genus *Apscaviroid*, is widely distributed in major apple-producing areas of China and several other countries and causes severe pome fruit diseases such as apple scar skin, dapple apple, and pear dimple [26]. The genomic sequence of ASSVd was first reported in 1987 and comprises nearly 330 nt [27]. ASSVd is seed-borne and persistent in infected apple trees [28]. Some studies reported that ASSVd is spread through whiteflies (Hemipterans) in greenhouses [29] but the spread mainly occurs through grafting or the use of contaminated equipment [30]. In the present study, we used ASSVd-infected apple trees to investigate the cross-infection of viroids to fungi and the consequences of the viroid infection of plants for plant-associated fungal communities. We examined the presence of ASSVd in fungal strains isolated from infected apple trees. Furthermore, we characterized the apple tree-associated fungal community profile (mycobiome) using high-throughput sequencing technology. We provided evidence that the transmission of ASSVd from plant to plant-associated fungi occurs in natural conditions, which may suggest the possible existence of mycoviroids in nature. Moreover, our results demonstrate the effects of viroid infection on the abundance of particular fungal species, indicating a possible role for viroids in the plant-fungus agroecosystem.

## 2. Materials and Methods

### 2.1. Collection of Plant Samples

The field sites where samples were collected are illustrated in Figure 1A. Apple fruits and twigs were collected from apple plant monoculture plots in Xi’an Guoyou Research Center, China Apple Agriculture Research System in Qianxian County, Shaanxi province, China, where apple plants (*Malus domestica Borkh.* cv. Qincui) have been grown since 2012. Apple trees are grown at intervals of 1.5 m in rows, with a 4 m distance between rows. All trees are maintained with the same agricultural managements. strategies. Samples were collected from apple trees in spring 2020 (at their median flowering time) and autumn 2019 (at their fruit maturing time). ASSVd infection was initially diagnosed by observing apple fruits and was further confirmed by reverse transcription-polymerase chain reaction (RT-PCR). Sequence analysis indicated that the ASSVd identified in this field is identical to the ASSVd KP1 isolate (GenBank accession no. MG602681) found in Korea [31]. A screening survey of ASSVd infection was conducted, and apple trees with or without ASSVd infection were identified. Three sites having ASSVd-infected or ASSVd-free trees were chosen for sample collection (Figure 1A). Nine ASSVd-infected and nine ASSVd-free trees were selected (three from each plot) for sample collection. Twenty fresh twigs collected from each tree were used for fungal isolation.

### 2.2. RNA Extraction, RT-PCR and Sequencing

Total RNA was extracted from apple twigs (2 cm in length) that had emerged a year prior, using TRIzol (Invitrogen, Waltham, MA, USA). Total RNA was also extracted from fungal mycelia cultured on potato dextrose agar (PDA; Becton, Dickinson and Co., Mountain View, CA, USA) with cellophane for 4–6 days following the previously described procedures [32]. The extracted RNA was quantified using a microvolume UV/Vis spectrophotometer (Implen NanoPhotometer, Westlake Village, CA, USA). For RT-PCR detection, first-strand cDNA was synthesized using ReverTra Ace reverse transcriptase (Toyobo, Osaka, Japan) and amplified using a 2× mixture of DNA polymerase (Kangwei, Guangzhou, China) with ASSVd-specific primers (Appendix A) for first and second nested PCR to detect viroid RNA. PCR products were subjected to Sanger sequencing.

### 2.3. DNA Extraction

The total DNAs of the plants and associated microbes were extracted from 10 twigs (0.5–1 cm in length) per tree using a DNA extraction kit (Omega Bio-tek, Norcross, GA, USA), and the resultant DNA extracts were evaluated via 1% agarose gel electrophoresis. The DNA concentrations were quantified using a NanoDrop2000 spectrophotometer (Thermo Scientific, Waltham, MA, USA) and normalized for use as the templates for generating PCR amplicons. Total DNA was extracted from fungal mycelia and cultured on PDA with cellophane using the phenol-based method as described previously [33].

### 2.4. Fungal Isolation and Single Spore Preparation

Apple twigs were cut into small pieces (roughly 0.2 × 0.5 cm) and sterilized with 75% alcohol. After washing with distilled water, the tissue samples were placed on a PDA plate containing streptomycin (50 μg/mL) to avoid bacterial contamination. Emerging fungal colonies were sub-cultured on a fresh PDA plate for further use. Fungal isolates were stored in a 10% glycerol solution and stored at −80 °C for long-term use. All fungal isolates were grown on PDA or Vogel’s medium [34] for 3–6 days at 24–26 °C for morphological observation or on cellophane-covered PDA plates for RNA and DNA extraction.

For single-spore isolation, ASSVd-carrying fungi were cultured on a benchtop for more than 4 weeks until asexual spores (conidia) developed. The spores were collected and distributed on a PDA plate at appropriate dilutions. Once the spores germinated, the mycelia were transferred to a new PDA plate and sub-cultured for morphological observation or RNA extraction and RT-PCR detection. The morphological comparison of ASSVd-carrying isolates and ASSVd-free isolates was performed with at least three independent fungal cultures.

### 2.5. Fungal Virulence Assay

The virulence of fungal isolates was assessed using fresh apple leaves collected from a healthy plant grown in a growth room. Fungi were inoculated on the sterilized leaves by placing a mycelial plug on a tiny wound made by a toothpick. After incubating on a benchtop (24–26 °C) for 5 to 7 days, the lesions were photographed and measured. All inoculations were repeated three times.

### 2.6. Fungal Species Identification

The extracted fungal DNA was used as a template for the PCR amplification of the intergenic transcribed spacer region of the nuclear ribosomal RNA genes (ITS1 and ITS4) [35] (Appendix A). The amplified ITS sequences were used as queries for BLASTn searches against GenBank standard databases (nt) or fungal ITS databases (Unite Release 8.0 http://unite.ut.ee/index.php (accessed on 9 May 2022).

### 2.7. Fungal Amplicon Sequencing 

Fungal communities were assessed based on the ITS [36] region of the eukaryotic ribosomal RNA gene using the primers ITS1F and ITS2R [37]. Briefly, PCRs were performed using 20 μL mixtures containing 10 ng of DNA template, 250 μM of dNTPs, 200 nM of primers, and 0.025 U of DNA polymerase (TransSart fastPfu, Tran, Beijing, China). The PCR procedure consisted of pre-denaturation at 95 °C for 3 min; 27 cycles at 95 °C for 30 s, 55 °C for 30 s, and 72 °C for 30 s; and stable extension at 72 °C for 10 min using the ABI Geneamp^®^ 9700 model thermal cycler (Thermo Fisher Scientific). Three PCR products per sample were pooled and recovered using a 2% agarose gel, purified using the Axyprep DNA Gel Extraction Kit (Axygen Biosciences, Union City, CA, USA), and qualified using the QuantusTM fluorometer (Promega, Madison, WI, USA). The purified PCR products were subjected to library construction using the NEXTFLEX@ Rapid DNA-Seq Kit according to the manufacturer’s instructions. The purified and qualified amplicon mixture was then sequenced using Illumina’s NovaSeq PE250 platform (Illumina, San Diego, CA, USA) performed by Shanghai Majorbio Bio-pharm Technology Co., Ltd. (Shanghai, China). Total DNA extracted from the same stem sample batches used for fungal isolation (collected in autumn 2019) was subjected to Illumina-based amplicon sequencing to analyze the fungal communities. To further investigate whether the viroid infection of apple trees alters the profile of plant-associated fungal communities, we repeated the Illumina-based amplicon sequencing to analyze the sequences of the collected stem samples from the same ASSVd-infected and ASSVd-free apple trees in the spring of the following year (2020, when the flowers blossomed). In this analysis, three ASSVd-infected and three ASSVd-free apple plants were analyzed independently, making a total of six analyzed samples.

### 2.8. Bioinformatics and Statistical Analyses

The raw sequencing data were analyzed using the Majorbio cloud platform (https://cloud.majorbio.com) (accessed on 9 May 2022). Briefly, the raw reads were quality-filtered using Fastp (https://github.com/OpenGene/fastp, version 0.19.6) (accessed on 9 May 2022) with the default settings [38]. The clean paired-end reads were merged as raw tags using Flash (http://www.cbcb.umd.edu/software/flash, version 1.2.11) (accessed on 9 May 2022) with a minimum overlap of 10 bp and a maximum mismatch ratio of 0.2 in the overlap region [39]. The effective tags were clustered into operational taxonomic units (OTUs) with a 97% similarity threshold using the UPARSE algorithm (http://drive5.com/uparse/, version 7.1) [40] (accessed on 9 May 2022). The OTUs were phylogenetically assigned against Unite Release 8.0 (http://unite.ut.ee/index.php) (accessed on 9 May 2022) for OTU taxonomic annotation using the RDP classifier (http://rdp.cme.msu.edu/, version 2.11) (accessed on 9 May 2022). The low-abundance reads of the OTUs (<0.01) were summed into “Others”. ITS function prediction analysis was performed using picrust2 (version 2.2.0) software [41]. Alpha diversities (Sobs, Shannon, Simpson, and Ace index) were calculated in Mothur (version1.30.2; https://www.mothur.org/wiki/Download_mothur) (accessed on 9 May 2022), and differences in the alpha diversities among ASSVd (−) and ASSVd (+) were calculated using *t*-tests. The beta diversities based on Bray–Curtis were calculated using QIIME (version 1.9.1; http://qiime.org/install/index.html) (accessed on 9 May 2022). The sequencing data have been deposited at NCBI (https://www.ncbi.nlm.nih.gov/sra) (accessed on 9 May 2022) under the accession number SRP394660.

The Wilcoxon rank-sum test was used to analyze the differences in alpha diversity between groups, and principal component analysis (PCA) based on the Bray–Curtis distance algorithm was used to test the similarities in microbial communities between samples. Pan analysis and the Sobs index at the OTU level were used to compare OTU richness between samples. Information on the specific analysis software is presented in Appendix A.

## 3. Results

### 3.1. The Presence of ASSVd in Fungal Isolates

The sequential experiments conducted in this study are illustrated in Figure 1A. ASSVd infection in the sampled apple trees was confirmed based on the appearance of disease symptoms on apple fruits (Figure 1B) and the RT-PCR assay (Figure 1C). To avoid the effect of environmental disparities on the phyllosphere fungal communities, plant samples were collected from an orchard with relatively homogeneous environmental conditions.

Fungal isolates were obtained from freshly collected and small-cut apple twigs (Figure 1D). In total, 117 fungal isolates were obtained from ASSVd-infected plants, while 301 fungal isolates were obtained from ASSVd-free plants (Figure 1E and Table 1). Based on ITS sequencing, 418 fungal isolates were classified into nine ascomycete species, including four *Alternaria* spp. (*A. tenuissima, A. compacta, A. alternata, and A. brassicicola)* from the *Pleosporaceae* family, *Epicoccum nigrum* from the *Didymellaceae* family, *Curvularia spicifera* from the *Pleosporaceae* family, *Talaromyces erruculosus* from the *Trichocomaceae* family, *Botryosphaeria dothidea* from the *Botryosphaeriaceae* family, and *Diaporthe phaseolorum* from the *Diaporthaceae* family; 14 isolates were unclassified.

The majority of fungal isolates obtained from ASSVd-infected plants were *A. alternata* (85.5%), with a lower proportion of *D. phaseolorum* (8.5%) and a few other species including unclassified fungi. The majority of the fungal isolates obtained from ASSVd-free plants were also *A. alternata* (63.8%) but a higher diversity of fungal species was observed, including some other *Alternaria* species such as *A. tenuissima*, *A. compacta,* and *A. brassicicola* (Table 1). The RT-PCR assay and confirmational sequencing of the PCR products was used to examine the 117 fungal isolates for the presence of ASSVd (Figure 1F and Appendix A). The results showed that 81 out of 117 (69.2%) of the fungal isolates carried ASSVd. The viroid was present in various species; 77 (77%) *A. alternate* isolates, two (50%) *E. nigurum* isolates, one (100%) *Botryosphaeria dothidea* isolate, and one (10%) *D. phaseolorum* isolate harbored ASSVd (Table 1). The analysis of the partial sequences of ASSVd genome showed nucleotide sequence differences between the viroid strains detected in the plant and fungal isolates (Appendix A), suggesting the presence of natural variants or the occurrence of viroid genome changes or adaptation in the fungal hosts, similar to the genome changes observed for HSVd and ASBVd in the fungal hosts [42].

### 3.2. Phenotypic Effects of ASSVd on Fungal Isolates

To examine whether ASSVd can cause fungal phenotypic changes, we generated ASSVd-free fungal isolates by single-spore isolation, usually carried out to eliminate mycoviruses (fungal viruses) from fungal strains [43,44]. Four ASSVd-carrying fungal isolates representing three different fungal species, *A. alternata, B. dothidea,* and *E. nigrum,* were subjected to repeated single-spore isolation as illustrated in Figure 2A. Since there are two distinct phenotypes among *A. alternata* isolates, two fungal strains (Q2-2 and Q2-3) were selected to represent each type (Figure 2B,D). Over 20 single-spore-germinated fungal isolates derived from every generation were tested by RT-PCR assay for the presence of ASSVd. After two rounds of single-spore isolation, ASSVd-free strains were obtained from the *B. dothidea* isolate (Q2-5G2-2), while ASSVd-free *A. alternata* and *E. nigrum* isolates (Q2-2G4-1, Q2-3G4-1 and Q2-1G4-3) were obtained after four rounds of single-spore isolation (Figure 2J). A comparison of fungal colony growth between ASSVd-infected and ASSVd-free strains showed that the presence of ASSVd markedly reduced the growth of *A. alternata* and *E. nigrum* on PDA medium while having no effect on *B. dothidea* growth (Figure 2B–I). Thus, ASSVd has different effects on fungal species when they are grown on a rich medium. In parallel, the same ASSVd-carrying and ASSVd-free strains were cultured in Vogel’s medium, a minimal medium for fungal growth [34]. In Vogel’s medium, ASSVd-carrying and ASSVd-free *A. alternata* and *E. nigrum* strains showed a similar growth and morphology, whereas the presence of ASSVd increased the growth of *B. dothidea* strains (Appendix A), indicating that ASSVd had no obvious negative effects on fungal growth in a minimal medium.

To investigate the effects of ASSVd on fungal pathogenicity, we inoculated the ASSVd-carrying and ASSVd-free fungal strains on fresh apple leaves. The ASSVd-carrying and ASSVd-free strains of *A. alternata* and *B. dothidea* caused the development of similarly sized lesions on apple leaves (Figure 2K–P), whereas ASSVd-carrying *E. nigrum* caused smaller lesions compared to the ASSVd-free strain (Figure 2Q,R), showing that ASSVd confers hypovirulence on *E. nigrum*.

### 3.3. Effect of ASSVd on the Composition of Plant-Associated Fungi

As ASSVd can spread to fungi and in some cases alter their phenotypes, we further investigated by high-throughput sequencing analysis whether ASSVd infection in apple trees affects the profile of the fungal communities associated with the trees in autumn 2019. A total of 290,178 (ASSVd(-):146, 892, and ASSVd(+):143,286) raw reads were obtained from Illumina MiSeq and yielded 145,089 (ASSVd(-):73,446, and ASSVd(+):71,643) high-quality fungal ITS reads after trimming and filtering. These high-quality sequences were clustered into 217 fungal OTUs (Appendix A). The dilution curves of each group based on the Sobs index at the OTU level (Appendix A) indicated that the majority of fungal species in the samples were present and the results reflected the fungal community. Variations in fungal composition were presented using a Venn diagram (Figure 3A). Overall, 112 (51.7%) OTUs overlapped between the ASSVd-infected and ASSVd-free stem samples, 59 (27.2%) OTUs were only present in ASSVd-free samples, and 46 (21.2%) OTUs were only present in ASSVd-infected stems (Figure 3A), suggesting that changes in fungal communities may occur due to viroid infection. The 10 most abundant fungal genera are listed in Appendix A. Compared to ASSVd-free stem samples, the relative abundances of five fungal genera were increased in ASSVd-infected stem samples, including four ascomycetes, *Cladosporium* (15.43% to 22.62%)*, Setomelanomma* (7.84% to 17.80%)*, Pyrenochaeta* (1.01% to 22.64%), and *Alternaria* (6.30% to 16.53%), and one basidiomycete *Filobasidium* (0.82% to 1.93%), whereas the relative abundances of other fungal genera were reduced, including *Erythrobasidium* (10.18% to 1.03%), *Leotionmycetes* (6.97% to 0.37%), *Neosetophoma* (3.34% to 0.43%), and genera in the family *Phaeosphaeriaceae* (32.69% to 8.26%; Figure 3B and Appendix A).

To further investigate whether the viroid infection of plants alters the profile of plant-associated fungal communities, we repeated our high-throughput sequencing analysis on collected stem samples from the same ASSVd-infected and ASSVd-free apple trees in the spring of the following year (2020, when the flowers blossomed). A total of 757,230 raw data reads were obtained from Illumina MiSeq, and 378,615 high-quality reads were obtained after trimming and filtering (Appendix A). The Sobs curves (Appendix A) of observed species richness (correlated with OTU richness) appeared to reach near-saturation, suggesting that the majority of fungal diversity was represented in all samples. Of a total of 195 OTUs, 130 (66.7%) OTUs, representing the majority of sequences, were observed in both ASSVd-infected and ASSVd-free samples. On the other hand, 25 (12.8%) unique OTUs were only present in ASSVd-free samples and 40 (20.5%) OTUs were specifically present in ASSVd-infected samples (Figure 3C). To investigate ASSVd-mediated fungal community changes, the *α*-diversity of the microbial community in each sample was estimated (Table 2). An analysis showed that Sobs, Shannon, Simpson, Ace, and Chao index values were not significantly different between the ASSVd-infected and ASSVd-free samples (Table 2). In *β*-diversity analysis, PCA showed that the fungal community structures did not significantly differ between ASSVd-infected and ASSVd-free samples (Figure 3D). Thus, the ASSVd infection of plants does not seem to largely affect the general profile of plant-associated fungal communities.

In the fungal communities, the 10 most abundant fungal groups (family, order, class, or genus) were Phaeosphaeriaceae, *Alternaria*, Pleosporales, *Cladosporium*, Didymellaceae, *Setomelanomma*, *Erythrobasidium*, *Chaetosphaeronema*, *Vishniacozyma*, and *Pyrenochaeta* (Figure 3E and Appendix A). Among them, Phaeosphaeriaceae, *Alternaria*, *Cladosporium, Setomelanomma, Erythrobasidium*, *Pyrenochaeta*, and Didymellaceae were consistently predominant in the fungal communities in both the spring 2020 and autumn 2019 samples. Notably, the genus Alternaria showed a consistent increase in abundance in ASSVd-carrying samples compared to the ASSVd-free samples in both spring 2020 (3.02–12.85% to 11.65–22.67%) and autumn 2019 (6.30% to 16.53%) (Figure 3B,E, Appendix A). The abundances of the fungal genera Cladosporium, Filobasidium, and Setomelanomma increased in the ASSVd-infected plant tissues in the autumn 2019 samples but not the spring 2020 samples (Figure 3B,E, Appendix A). Thus, the ASSVd infection of apple trees appears to specifically affect the abundance of the Alternaria species that colonize the trees.

## 4. Discussion

Although existing as the smallest known infectious agents with highly base-paired, single-stranded noncoding RNAs, viroids often cause disease symptoms in plant hosts [45,46]. To date, nearly 30 plant diseases are known to be associated with viroids, primarily vegetables, field and ornamental crops, fruit and palm trees, and grapevines [4,5,36,46]. The modes of viroid transmission have been recently reported, including mycoviroid transmission [47]. Previously, under laboratory conditions, we demonstrated that viroids could be transmitted between plants and co-colonizing phytopathogenic fungi [15]. Similarly, in the present study, we extended our previous analysis of viroid cross-kingdom transmission to assess the spread of viroids to plant-associated fungi under natural conditions. For this purpose, we selected ASSVd and its natural host, the apple tree, as the plant-viroid-fungus ecosystem under study. After confirming the presence of ASSVd, a large number of filamentous fungi were isolated from viroid-infected trees. A total of 69% of the isolated fungal isolates tested positive for ASSVd and some strains maintained ASSVd during several rounds of subculturing, indicating that the viroid can spread from plants to the plant-associated filamentous fungi. These results were consistent with the transmission of viroid to fungus demonstrated under laboratory conditions [15], suggesting that viroids can spread to plant-associated fungi during the colonization of a plant under natural conditions. In addition, these results also propose the hypothesis of a new mode of viroid spread in nature, that is, the involvement of plant-associated fungi in viroid transmission.

Plants often simultaneously harbor a variety of fungi and viruses [48,49]. Our previous works also demonstrated that viruses were bi-directionally transferred between plants and pathogenic fungi, suggesting the transfer of virus particles or viral RNA between plants and fungi [50,51,52]. As cellular agents, viruses or viroids are possibly secreted into the extracellular space, where they are ingested by fungi. Interestingly, a recent study revealed that small RNAs and long noncoding RNAs (IncRNAs) including circular RNAs, are rich in the apoplastic wash fluid (AWF) purified from the extracellular spaces of *Arabidopsis* leaves [53]. Two plant RNA binding proteins, glycine-rich RNA-binding protein 7 (GRP7) and a small RNA-binding protein argonaute 2 (AGO2) were found to co-immunoprecipitate with IncRNAs (150–500 nt). The mutation of *GRP7* and *AGO2* remarkably reduces the abundance of IncRNAs in AWF, suggesting that these RNA binding proteins contribute to IncRNA secretion or stabilization [53]. Hence, viroids, noncoding, circular RNAs, probably enter the extracellular spaces via a similar pathway. It would be interesting to study the cellular proteins associated with viroid cross-kingdom transmission to plants in the future.

A large number of mycoviruses have been discovered in a variety of fungal species, including plant pathogenic fungi [44,54]. Mycoviruses could influence their hosts [54,55,56]. Previously, we found that HSVd could stably replicate in ascomycete filamentous fungi and cause disease hypovirulence-associated symptoms on the phytopathogenic fungus *V. mali* but not on two other fungi, *F. graminearum* and *C. parasitica* [15]. Our current study revealed that various fungal species could host ASSVd, including *A. alternata*, *B. dothidea*, and *E. nigrum*, which are all known as plant pathogens and endophyte fungi in the phylum Ascomycota. Thus, these findings further support our proposal regarding the possible existence of mycoviroids in nature [15,47]. Relative to ASSVd-free fungal isolates, ASSVd-carrying isolates of *A. alternata* and *E. nigrum* exhibited slightly reduced growth on a PDA medium. However, only the pathogenicity of *E. nigrum* appeared to be affected by the presence of ASSVd. This study provides further evidence that the horizontal transfer of plant viroids to fungi could induce symptoms in certain filamentous fungi.

As ASSVd can cause phenotypic changes in both plant and fungal hosts, ASSVd can further affect the interactions of plants and plant-associated fungi. Our analysis showed that the ASSVd infection of apple trees does not greatly affect the diversity and structure of plant-associated fungal communities; however, ASSVd infection affects the abundance of *Alternaria* species. The ASSVd infection of apple trees increased the abundance of *Alternaria* while decreasing the abundance of species from the family Phaeosphaeriaceae. Many fungi in the family Phaeosphaeriaceae are necrotrophic or saprobic [57], while *Alternaria* species associated with apple plants are known as major plant pathogens [58,59]. These results suggest that ASSVd-infected apple trees hosted a higher proportion of pathogenic fungi. The mechanism by which ASSVd affects *Alternaria* and enhances its richness in fungal communities is still unclear. It might be that the spread of ASSVd to or cross-infection in *Alternaria* and other fungal species alters the composition of fungal communities due to the effects of ASSVd on fungal growth and physiology. Another possibility is that ASSVd indirectly affects *A. alternata* propagation in the plants. Plant disease conditions could shape the plant microbiome [60,61,62]. Plant pathogens or invading organisms can regulate host immune responses. ASSVd induced disease symptoms, such as scar skin on the apple fruits of infected apple trees, and thus possibly affected the plant’s immune responses. Moreover, recent works highlighted the role of the plant immune system in microbiome assembly. Microbe-triggered immunity, known as microbe-associated molecular patterns (MAMPs), affects microbe proliferation [62,63,64]. ASSVd possibly suppresses plant immunity and promotes the expansion of plant pathogens, resulting in the enrichment of certain pathogens such as *A. alternata*. Our results indicate complex triple interactions between plants, fungi, and viroids that are of interest for future studies.

## Figures and Tables

**Figure 1 cells-11-03686-f001:**
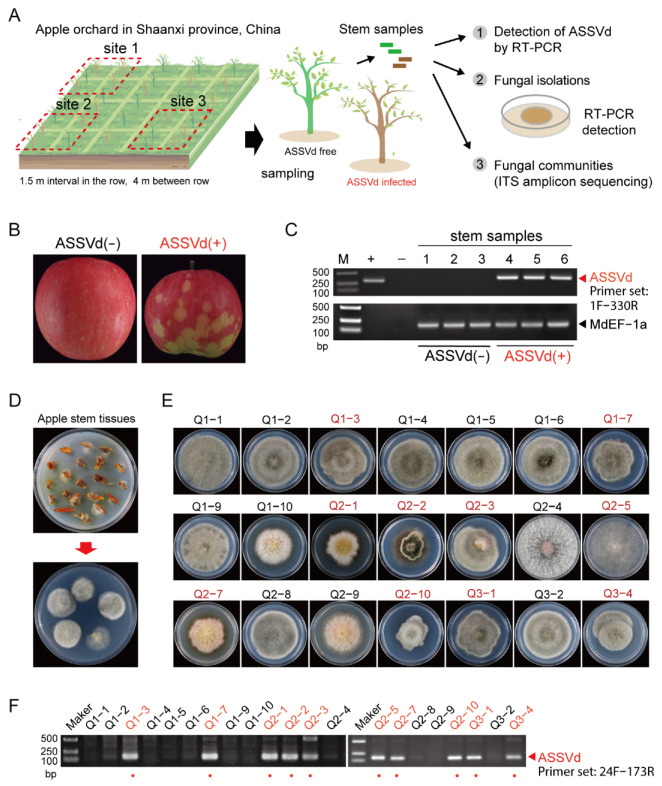
Assessment of ASSVd spread from apple trees to tree-associated fungi under natural conditions. (**A**) Experimental procedures carried out in this study. (1) After ASSVd symptom diagnosis and RT-PCR detection, apple trees were selected from three sites in one orchard under the same natural conditions. (2) Stem samples were collected from the selected viroid-infected or viroid-free apple plants, and plant-associated fungi were isolated from the samples and subjected to detection of ASSVd infection by RT-PCR. (3) Fungal amplicon sequencing was carried out to analyze the fungal communities present in ASSVd-infected and ASSVd-free apple stem tissues. (**B**) ASSVd induced disease symptoms in apple fruits harvested from the tested apple tree. (**C**) Detection of ASSVd in apple trees by RT-PCR. The host gene *Apple ELONGATION FACTOR1a* (*MdEF-1a,* DQ341381) was used as a control to verify the quality of the RNA. “M” indicates the DNA ladder, “+” indicates the positive control comprising a plasmid carrying an ASSVd cDNA fragment used as the PCR template, and “−” indicates the negative control without RT reaction. (**D**) The stem tissues were placed on PDA medium for fungal isolation. (**E**) The colony morphology of representative fungal isolates on PDA medium. (**F**) The detection of ASSVd by RT-PCR in the fungal isolates shown in (**E**).

**Figure 2 cells-11-03686-f002:**
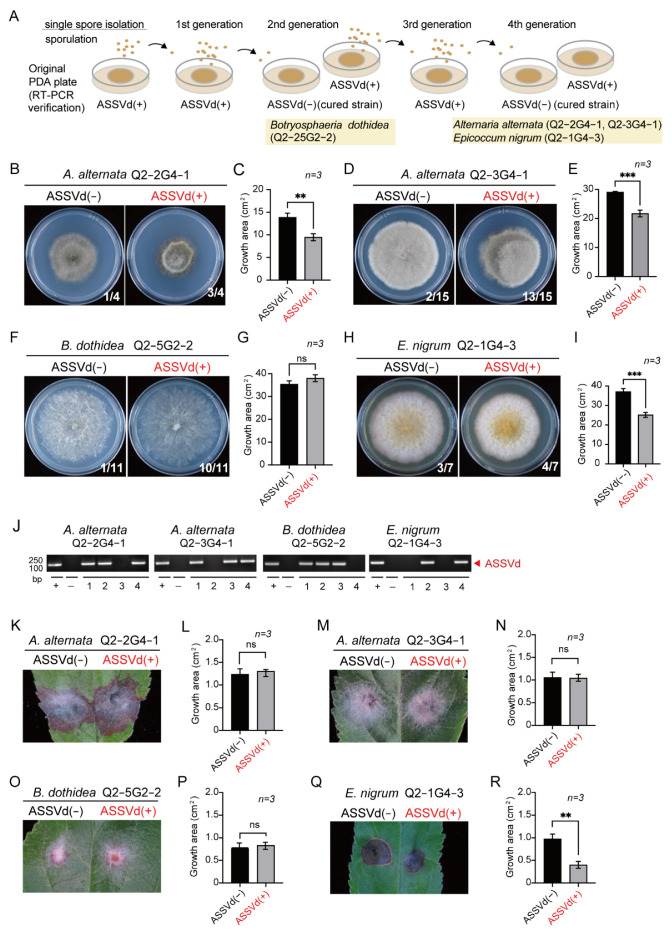
Effects of ASSVd infection on fungal isolates. (**A**) Experimental procedure for obtaining ASSVd-free isolates from ASSVd-carrying isolates. (**B**,**D**,**F**,**H**) Phenotypic growth of ASSVd-carrying and ASSVd-free fungal isolates. All isolates were grown on PDA medium (6 cm plate) for 5–6 days and photographed. ASSVd (+) and ASSVd (−) indicate carrying and free isolates, respectively. (**C**,**E**,**G**,**I**) Measurement of the growth area of fungal colonies. Data are presented as mean ± SD (*n* = 3). **, and *** indicate significant differences at *p* < 0.01, and 0.001, respectively (Student’s *t*-test). (**J**) RT-PCR detection of ASSVd accumulation in the fungal isolates germinated from single spores described in (**A**). (**K**,**M**,**O**,**Q**) Fungal virulence assay on apple leaves. Fresh apple leaves were inoculated with mycelial plugs and photographed 5 days later. (**L**,**N**,**P**,**R**) The lesion area measured on the inoculated apple leaves described in C, E, G, and I. ** indicate a significant difference at *p* < 0.01 (Student’s *t*-test).

**Figure 3 cells-11-03686-f003:**
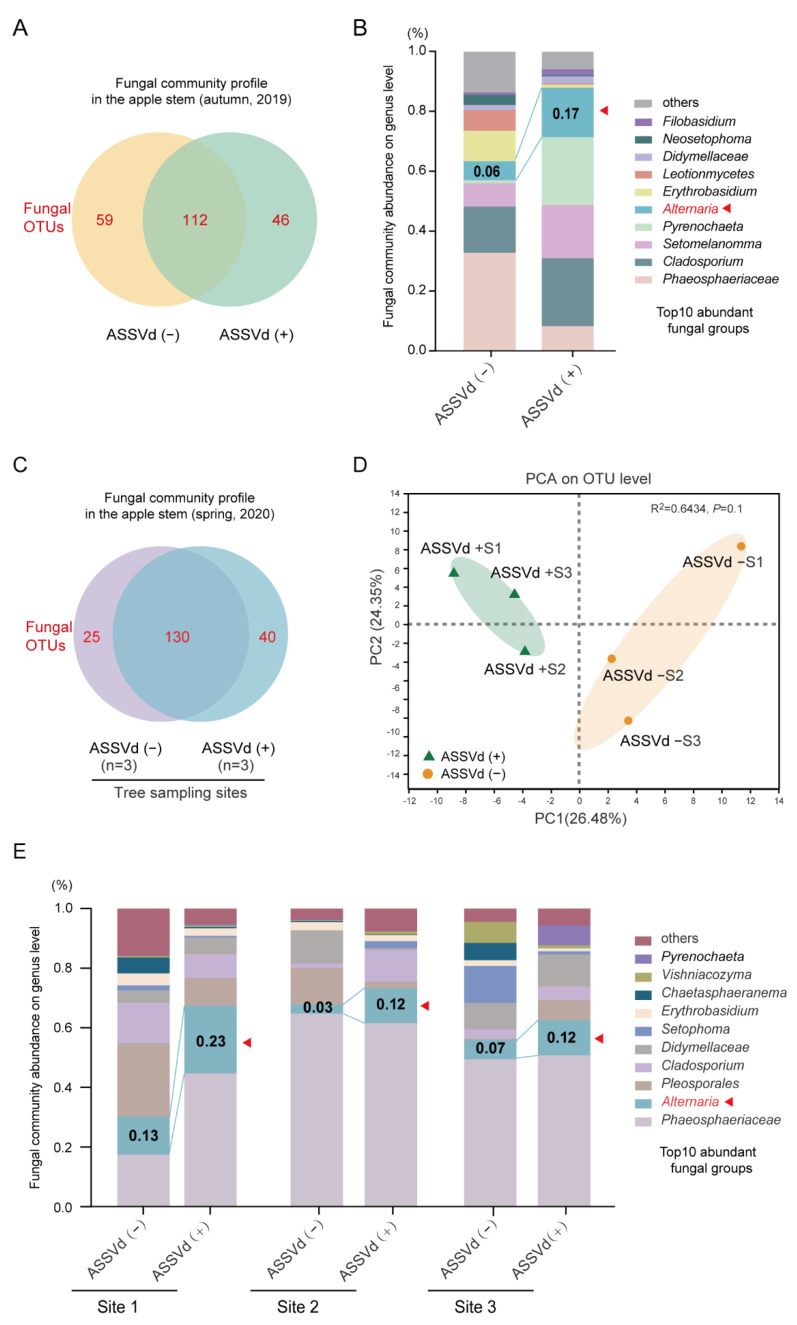
Fungal community profile in ASSVd-infected and ASSVd-free apple stem tissues. (**A**) Similarities and differences in fungal OTUs between two samples collected in the autumn of 2019 presented as a Venn diagram. (**B**) Composition of the fungal communities in the two samples collected in the autumn of 2019 classified at the genus level. The 10 most abundant fungal genera are presented. (**C**) Similarities and differences in fungal OTUs between two samples collected in the spring of 2020 presented as a Venn diagram. (**D**) Principal component analysis (PCA) of two samples collected in the spring of 2020 based on Bray–Curtis distances. (**E**) Composition of the fungal communities in the two samples collected in the spring of 2020 classified at the genus level. The 10 most abundant fungal genera are presented.

**Table 1 cells-11-03686-t001:** List of fungal isolates obtained from ASSVd-infected and ASSVd-free apple stem samples collected in the autumn of 2019.

Fungal Genus	Fungi Isolated from ASSVd-Infected Plant Tissues	Fungi Isolated from Non-Infected Plant Tissues
	No. Infected/No. Tested Strains	No. Strains
*Alternaria tenuissima*	0	7
*Alternaria compacta*	0	42
** *Alternaria alternata* **	**77/100**	**192**
*Alternaria brassicicola*	0	10
** *Epicoccum nigrum* **	**2/4**	**15**
*Curvularia spicifera*	0	1
*Talaromyces verruculosus*	0	5
** *Botryosphaeria dothidea* **	**1/1**	**2**
** *Diaporthe phaseolorum* **	**1/10**	**13**
Unclassified	0/2	14
Total	81/117	301

**Table 2 cells-11-03686-t002:** Alpha diversity of the fungal communities present in ASSVd-infected and ASSVd-free apple stem samples collected in the spring of 2020.

Sample	Sobs	Shannon	Simpson	Ace
ASSVd (−)	112.67 ± 9.07 a	2.16 ± 0.45 a	0.23 ± 0.10 a	139.65 ± 13.94 a
ASSVd (+)	122.67 ± 1.53 a	2.17 ± 0.26 a	0.21 ± 0.10 a	151.52 ± 5.78 a

Values in the table are the mean ± SD (*n* = 3). The same letters after values indicate non-significant difference for the data between the same column at *p* ≤ 0.05 (Student’s *t*-test).

## Data Availability

Data are available from corresponding author upon reasonable request.

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
