# Peer review of "Natural Cross-Kingdom Spread of Apple Scar Skin Viroid from Apple Trees to Fungi"

_cells, 2022, doi:10.3390/cells11223686_

Round 1

Reviewer 1 Report

Authors Mengyuan et al. presented here manuscript entitled „Natural cross-kingdom spread of apple scar skin viroid from apple trees to fungi“.

Authors made a great work on analysis of viroid transfer from apple trees into fungi.

During the work, authors analysed viroid sequences from apple trees and their fungal inhabitants. They also analysed differencies of fungal spectrum present in viroid-infected and viroid-free apple trees.

Authors described differencies in properties of viroid-infected and viroid-free fungal cultures inluding their pathogenicity for original hosts, e.g. apple trees.

I also appreciate high quality images of gels and fungal cultures.

One comment - please mention the sequencing of viroids also in the Material and Methods section.

Final recommendation – accept.

Author Response

Reviewer 1

Authors Mengyuan et al. presented here manuscript entitled „Natural cross-kingdom spread of apple scar skin viroid from apple trees to fungi“.

Authors made a great work on analysis of viroid transfer from apple trees into fungi.

During the work, authors analysed viroid sequences from apple trees and their fungal inhabitants. They also analysed differencies of fungal spectrum present in viroid-infected and viroid-free apple trees.

Authors described differencies in properties of viroid-infected and viroid-free fungal cultures inluding their pathogenicity for original hosts, e.g. apple trees.

I also appreciate high quality images of gels and fungal cultures.

One comment - please mention the sequencing of viroids also in the Material and Methods section.

Final recommendation – accept.

  • Answer: Thank you very much for your interest and positive comments on our work. We have added the information on viroid sequencing.

Reviewer 2 Report

The manuscript (ID 2013520) submitted to the ‘Cells’ Journal entitled “Natural Cross-Kingdom Spread of Apple Scar Skin Viroid from Apple Trees to Fungi” is an informative study and produced exciting results. The findings of the publication indicate the spread of ASSVd from apple plants (higher kingdom) to fungi (lower kingdom) under natural conditions including A. alternata as a major host and further suggest the possible existence of mycoviroids in nature. They also elucidate that ASSVd affected the growth and pathogenicity of certain fungal species and also confer hypovirulence on an ascomycete, Epicoccum nigrum. This study provides further evidence that viroid infection could induce symptoms in certain filamentous fungi. Mycobiome analyses suggest that ASSVd infection did not generally affect the diversity and structure of fungal communities but specifically increased the abundance of Alternaria species. However, the authors of MS are requested to provide some justifications as mentioned below and the manuscript will be worth publishing after some rectifications:

1)       The manuscript should be thoroughly checked for grammatical corrections, sentence formation, and repetition of words. Delete the old reference if a recent one has been provided.

2)       Line-16: delete “naturally”.

3)       Line-18: add naturally after ASSVd.

4)       Line-21: Add “an” before “ascomycete”. Delete “isolates”.

5)       Line-22-23: replace “At least-----fungal species” with “Out of the examined samples, viroid was detected in 81 isolates (69.2%) including A. alternata as well as other fungal species”.

6)       Line-23-25: delete “To examine------spore isolation”.

7)       Line-25: Add “developed by single spore isolation” after “ASSVd-free”.

8)       Line-39: delete reference no. 2.

9)       Line-45: reference no. 3 need to replace by 8,9.

10)    Line-71: (19-23), keep only the most relevant recent references.

11)    Line-114: add “Assessment of” before ASSVd spread.

12)    Line-120: add “induced” between ASSVd and disease.

13)    Line-129: delete “1-year-old” and add “that were emerged a year ago” after “apple twigs” and mention the size taken also.

14)    Line-130: replace “in” with “on”.

15)    Line-162-163: replace “The virulence of -------- growth room” with “The virulence of fungal isolates was assessed using fresh apple leaves collected from a healthy plant grown in a growth room”.

16)    Line-224: replace “ASSVd infection….for sampling” with “ASSVd infection in sampled apple trees”.

17)    Line-256-276: Have the authors performed study of phenotypic effects of ASSVd on fungal isolates using only fungal spores of A. alternate, B. dothidea, and E. nigrum for both the ASSVd-free and infected isolates. As in the case of obtaining ASSVd-free fungal isolates, they obtained ASSVd-free fungal spores first and then carried out a comparative study on PDA. Similarly, have they used ASSVd-infected fungal spores?

18)    Line-311-312: Use big and small brackets to clear the reads mentioned.

Author Response

Reviewer 2

The manuscript (ID 2013520) submitted to the ‘Cells’ Journal entitled “Natural Cross-Kingdom Spread of Apple Scar Skin Viroid from Apple Trees to Fungi” is an informative study and produced exciting results. The findings of the publication indicate the spread of ASSVd from apple plants (higher kingdom) to fungi (lower kingdom) under natural conditions including A. alternata as a major host and further suggest the possible existence of mycoviroids in nature. They also elucidate that ASSVd affected the growth and pathogenicity of certain fungal species and also confer hypovirulence on an ascomycete, Epicoccum nigrum. This study provides further evidence that viroid infection could induce symptoms in certain filamentous fungi. Mycobiome analyses suggest that ASSVd infection did not generally affect the diversity and structure of fungal communities but specifically increased the abundance of Alternaria species. However, the authors of MS are requested to provide some justifications as mentioned below and the manuscript will be worth publishing after some rectifications:

  • Answer: Thank you for your time to review our article. We greatly appreciate your positive comments and suggestions on our manuscript.

1)       The manuscript should be thoroughly checked for grammatical corrections, sentence formation, and repetition of words. Delete the old reference if a recent one has been provided.

- Answer: We have checked the revised manuscript thoroughly.

2)       Line-16: delete “naturally”.

- Answer: The word has been deleted

3)       Line-18: add naturally after ASSVd.

- Answers: The word has been added

4)       Line-21: Add “an” before “ascomycete”. Delete “isolates”.

- Answers: “an” has been added.

5)       Line-22-23: replace “At least-----fungal species” with “Out of the examined samples, viroid was detected in 81 isolates (69.2%) including A. alternata as well as other fungal species”.

- Answers: As suggested, the sentence has been changed.

6)       Line-23-25: delete “To examine------spore isolation”.

- Answers: The sentence has been deleted.

7)       Line-25: Add “developed by single spore isolation” after “ASSVd-free”.

- Answers: The texts have been added.

8)       Line-39: delete reference no. 2.

- Answers: Reference has been removed.

9)       Line-45: reference no. 3 need to replace by 8,9.

- Answers: Reference has been replaced

10)    Line-71: (19-23), keep only the most relevant recent references.

- Answers: Two references have been removed.

11)    Line-114: add “Assessment of” before ASSVd spread.

- Answers: The texts have been added.

12)    Line-120: add “induced” between ASSVd and disease.

- Answers: The text has been added.

13)    Line-129: delete “1-year-old” and add “that were emerged a year ago” after “apple twigs” and mention the size taken also.

- Answers: The sentence has been modified as suggested.

14)    Line-130: replace “in” with “on”.

- Answers: The text has been replaced.

15)    Line-162-163: replace “The virulence of -------- growth room” with “The virulence of fungal isolates was assessed using fresh apple leaves collected from a healthy plant grown in a growth room”.

- Answers: Sentences have been modified, as suggested.

16)    Line-224: replace “ASSVd infection….for sampling” with “ASSVd infection in sampled apple trees”.

- Answers: The sentence has been replaced.

17)    Line-256-276: Have the authors performed study of phenotypic effects of ASSVd on fungal isolates using only fungal spores of A. alternate, B. dothidea, and E. nigrum for both the ASSVd-free and infected isolates. As in the case of obtaining ASSVd-free fungal isolates, they obtained ASSVd-free fungal spores first and then carried out a comparative study on PDA. Similarly, have they used ASSVd-infected fungal spores?

- Answers: Assuming that we do not misunderstand your question; we carried out single-spore isolations from ASSVd-infected A. alternate, B. dothidea, and E. nigrum and then compared the phenotype of the fungal progenies generated from spores that have been confirmed to carry ASSVd or to are free from ASSVd.

18)    Line-311-312: Use big and small brackets to clear the reads mentioned.

- Answers: Brackets have been changed.

Reviewer 3 Report

High quality work which done after a lot of experiments, a very clear Discussion, merit to be accepted,Its good work which demonstrate a huge data collected and analysed.

1-Regarding line 250 "ASSVd genome  showed nucleotide sequence differences between the viroid strains detected in the plant and fungal isolates" i believe that verifying the pathogencity of ASSVd from isolates by  re-inoculation on plants will  be much interesting. Line 270:I think here should be added letter F for B.dothidea

 2-It will be better to investigate the  titer & localization of viroid  in fungal cells by gold-labeled in situ hybridization by light and transmission electron microscopy as been done  in this work for plant viruses infect fungi 

https://www.pnas.org/doi/full/10.1073/pnas.1315668111

3-Line287:I think the letters here should be revised

4-Line 304:I think the letter P should be O

5-Line 305:I think The Q should be U

Author Response

Reviewer 3

High quality work which done after a lot of experiments, a very clear Discussion, merit to be accepted, Its good work which demonstrate a huge data collected and analysed.

  • Answer: Thank you for time to review our article. We greatly appreciate your positive comments and suggestions on our manuscript.

1-Regarding line 250 "ASSVd genome  showed nucleotide sequence differences between the viroid strains detected in the plant and fungal isolates" i believe that verifying the pathogencity of ASSVd from isolates by  re-inoculation on plants will  be much interesting.

Answer: It is a good suggestion. In the future study. we will re-inoculate the ASSVd strains that were derived from fungi to plants.

Line 270:I think here should be added letter F for B.dothidea.

Answer: We have mentioned “Figure 2B-I” that means including “F” for B.dothidea.

 2-It will be better to investigate the  titer & localization of viroid  in fungal cells by gold-labeled in situ hybridization by light and transmission electron microscopy as been done  in this work for plant viruses infect fungi

https://www.pnas.org/doi/full/10.1073/pnas.1315668111

Answer: Viroids are lacking protein coat, therefore, it cannot be observed by immune-labelled transmission electron microscopy. Moreover, very low titer of viroid accumulation in fungi, so that it can not be detected by northern blot, that why we only detected ASSVd by RT-PCR. This is also the reason why we did not detect ASSVd using In situ hybridization.

3-Line287: I think the letters here should be revised

Answer: Letters have been revised.

4-Line 304:I think the letter P should be O

Answer: Measurement graph for B.dothidea (P) is included in this description

5-Line 305: I think The Q should be U

Answer: Thank you for pointing this mistake. Q should be labelled instead of U in the Figure 2. Correction has been done.
